# Ethanol Extract of *Amomum tsao-ko* Ameliorates Ovariectomy-Induced Trabecular Loss and Fat Accumulation

**DOI:** 10.3390/molecules26040784

**Published:** 2021-02-03

**Authors:** Ki-Shuk Shim, Youn-Hwan Hwang, Seon-A Jang, Taesoo Kim, Hyunil Ha

**Affiliations:** 1Herbal Medicine Research Division, Korea Institute of Oriental Medicine, Yuseong-daero 1672, Yuseong-gu, Daejeon 34054, Korea; angeloshim@kiom.re.kr (K.-S.S.); hyhhwang@kiom.re.kr (Y.-H.H.); white7068@kiom.re.kr (S.-A.J.); xotn91@kiom.re.kr (T.K.); 2University of Science & Technology (UST), Korean Convergence Medicine Major KIOM, 1672 Yuseongdae-ro, Yuseong-gu, Daejeon 34054, Korea

**Keywords:** *Amomum tsao-ko*, osteoporosis, ovariectomy, osteoclast differentiation

## Abstract

In Asia, *Amomum tsao-ko* has long been used as a spice or seasoning in food to stimulate digestion. In the present study, we evaluated the effects of ethanol extract of *Amomum tsao-ko* (EEAT) on menopausal osteoporosis and obesity. After the administration of EEAT in ovariectomy (OVX) mice models for five weeks, microcomputed tomography and a histological analysis were performed to assess, respectively, the trabecular structure and the fat accumulation in adipose, liver, and bone tissues. We also examined the effects of EEAT on a bone marrow macrophage model of osteoclastogenesis by in vitro stimulation from the receptor activator of nuclear factor-kappa Β ligand (RANKL) through real-time PCR and Western blot analysis. In addition, ultrahigh performance liquid chromatography–tandem mass spectrometry (UHPLC–MS/MS) with authentic standards was applied to characterize the phytochemical profiling of EEAT. We found that EEAT significantly decreased OVX-induced body weight gain and fat accumulation, significantly prevented OVX-induced deterioration of bone mineral density and microstructure of trabecular tissues, and significantly inhibited osteoclast differentiation by downregulating NF-κB/Fos/NFATc1 signaling in osteoclasts. Furthermore, UHPLC–MS/MS identified eight beneficial phytochemicals in EEAT. Collectively, these results suggest that EEAT might be an effective nutraceutical candidate to attenuate menopausal osteoporosis by inhibiting osteoclastogenesis and to prevent obesity by suppressing fat accumulation.

## 1. Introduction

Menopause is the natural cessation of menstruation due to the loss of ovarian function. It is accompanied by a high prevalence of severe physiological complications, including osteoporosis and obesity, two factors which have led to considerable clinical and public interest in the subject [1,2]. Estrogen regulates bone homeostasis by suppressing osteoclastic bone resorption and supporting osteoblast and osteocyte bone formation. When estrogen levels decline, osteoclastic bone resorption rapidly increases, which results in an imbalance of bone remodeling, weakness of bone microstructure, and net bone loss. Hormone treatment in animal ovariectomy (OVX) models or menopausal women by estradiol injection or supplementation of estrogen nutraceuticals restores bone turnover by inhibiting bone resorption while maintaining bone formation [3,4,5]. Estrogen also regulates osteoclast differentiation by increasing the decoy receptor of receptor activator of nuclear factor-kappa Β ligand (RANKL), or by increasing osteoclast apoptotic cell death [6]. In addition to its role in bone remodeling, estrogen affects energy metabolism during menopause. Estrogen is shown to exert a regulatory effect on food intake and energy expenditure by interacting with neuropeptides in rodent models [7]. It has been reported that estrogen loss could increase glucose intolerance, insulin resistance, and fat redistribution to the intra-abdominal area, all of which lead to metabolic complications associated with obesity in women [8,9].

Recently, nutraceuticals have gained increasing public interest due to their potential to relieve menopausal symptoms, osteoporosis, and obesity [10]. *Amomum tsao-ko*, known as black cardamom, is the dry pod of *Amomum subulatum* belonging to the family Zingiberaceae. It is widely used as a spice or seasoning in food and as a traditional medicinal herb to treat indigestion, stomach pain, or malaria [11,12]. *Amomum tsao-ko* extracts and several active components, including catechin, have antioxidant, anti-inflammatory, and antifungal properties [13,14,15]. Recently, *Amomum tsao-ko* was shown to suppress fat accumulation in the liver of a rat model fed with a high-fat diet, and to suppress hyperlipidemia in a rabbit model fed with high cholesterol [16,17]. These beneficial effects on obesity could be due to its antioxidative activity, antilipase activity, or regulatory role in lipid metabolism. In addition, an ethanol extract of *Amomum tsao-ko* (EEAT) was shown to significantly inhibit adipogenesis and reduce PPARγ and C/EBPα expression in 3T3-L1 cells in a dose-dependent manner [18]. It has also been suggested that obesity is associated with bone impairment by spontaneously enhanced osteoclastogenesis, which could be inhibited by nutraceuticals [19]. However, the pharmacological effects of *Amomum tsao-ko* on estrogen loss-induced obesity, osteoporosis, or osteoclastogenesis remain unknown.

To evaluate the pharmacological effect of *Amomum tsao-ko* on osteoporosis as well as obesity, we investigated the effect of EEAT on OVX-induced bone loss and obesity. The ovariectomized model is a gold standard model of menopausal osteoporosis exhibiting OVX-induced body weight gain [8]. In this study, the effect of EEAT on OVX-induced bone structural deterioration was evaluated by microcomputed tomography (micro-CT) analysis. Fat accumulation in the adipose, liver, and bone marrow tissue was determined by hematoxylin and eosin staining and histological analysis. Therefore, we examined the effect of EEAT on RANKL-induced osteoclastogenesis and RANKL signaling using bone marrow-derived macrophage cells (BMMs). To identify its phytochemical components, EEAT was characterized by ultrahigh performance liquid chromatography–tandem mass spectrometry (UHPLC–MS/MS) with a comparison of absorbance and mass fragmentation of standard marker components.

## 2. Results and Discussion

### 2.1. EEAT Inhibits RANKL Signaling during Osteoclast Differentiation

RANKL or RANK induces the phenotypical change of monocyte precursor cells into multinuclear osteoclasts, representing high levels of tartrate-resistant acid phosphatase (TRAP) activity and staining. We first investigated the effect of EEAT on RANKL-induced osteoclastogenesis using a TRAP assay. EEAT significantly reduced the numbers of TRAP-stained osteoclasts in a dose-dependent manner, as shown in Figure 1A,B. At the same concertation range, Figure 1C shows that 100 ug/mL EEAT decreased cell viability, suggesting that the maximum inhibitory concentration of EEAT on osteoclastogenesis was 33.3 ug/mL without any toxic effect. When mature osteoclasts were seeded on artificial bone slices to evaluate resorbing activity, EEAT did not inhibit pit formation, even at a concentration of 33.3 ug/mL; see Figure 1D. These findings indicate that EEAT specifically impairs osteoclast differentiation but not osteoclast activity. Osteoclastogenesis involves RANKL-activated signaling cascades to induce osteoclast-specific gene expression. Fos and nuclear factor of activated T-cells cytoplasmic 1 (NFATc1) are key transcription factors that drive osteoclastogenesis at an early stage [20]. However, their target genes, dendritic cell–specific transmembrane protein (DC-STAMP) and ATPv0d2, are responsible for the fusion of monocytes to multinuclear osteoclasts [21,22]. We found that EEAT significantly inhibited mRNA and protein expression of the transcription factors, shown in Figure 2A, as well as mRNA expression of the target genes, shown in Figure 2B. To elucidate the mechanism of EEAT action, we further examined the RANKL-induced upstream signaling pathway of Fos and NFATc1. Canonical NF-κB signaling regulates Fos expression, which subsequently stimulates NFATc1 to form a transcriptional complex during osteoclastogenesis [23]. Figure 2C shows that EEAT inhibited IκBα phosphorylation and degradation, and partially suppressed p65 and JNK phosphorylation. These results suggest that EEAT specifically inhibits RANKL-induced signaling and the NF-κB/Fos/NFATc1 pathway, which could result in retardation of osteoclast differentiation from precursor cells.

### 2.2. EEAT Ameliorates Bone Loss in OVX Mice

Estrogen loss stimulates osteoclast differentiation and bone resorption activity at a higher rate than that of bone formation in the early phase of menopausal osteoporosis, which further accelerates the turnover rate of bone remodeling to increase osteoporotic bone loss [24,25]. In the micro-CT image from the OVX group, we observed impaired trabecular bone quality with a decrease in trabecular volume and number, which was confirmed by five bone structure parameters shown in Figure 3A: bone mineral density (BMD), trabecular bone volume fraction (BV/TV), trabecular number (Tb.N), trabecular separation (Tb.Sp), and trabecular thickness (Tb.Th). When compared with the sham group, OVX decreased BMD (44%), BV/TV (50%), Tb.N (41%), and Tb.Th (13%) but increased Tb.Sp (92%). A high dose of EEAT (100 mg/kg) significantly improved these parameters by increasing BMD (25%), BV/TV (55%), Tb.N (36%), and Tb.Th (12%) and decreasing Tb.Sp (26%), compared with the OVX group. To further investigate the effect of EEAT on bone metabolism, serum bone turnover markers were measured. EEAT decreased both the levels of C-terminal telopeptide of type I collagen (CTX), a bone resorption marker, and procollagen type I N-terminal propeptide (PINP), a bone formation marker, compared with the OVX group, shown in Figure 3B. Therefore, these results suggest that improvement of the bone structure parameters in the OVX model by EEAT administration could have resulted from its effect to decrease the turnover rate of bone remodeling at least in part by suppressing osteoclast-mediated bone resorption. Low concentration of EEAT (30 mg/kg) exhibited a similar degree of inhibition of deterioration of bone parameters, suggesting the efficiency of EEAT, even at low concentration, to prevent OVX-induced osteoporosis.

### 2.3. EEAT Inhibits Fat Accumulation in OVX Mice

In addition to menopausal bone loss, the OVX model represents a useful model of menopausal obesity. The model effectively depicts altered energy balance and fat distribution, which result in an increase in body weight and obesity [26,27]. Estrogen deficiency was revealed by a decrease in uterine weight, which resulted in an increase in body weight and gonadal fat, compared with the sham group; see Figure 4A–C. EEAT administration inhibited the increase in body weight in a dose-dependent manner without changing the uterine weight, suggesting the non-estrogenic activity of EEAT. To clarify the effect of EEAT on fat storage, hematoxylin and eosin staining of liver, fat, and bone tissues was performed. As seen in Figure 4D, OVX mice showed significant enlargement of adipocytes in adipose tissue and accumulation of numerous lipid droplets in non-adipose tissues (liver and bone). However, a high dose of EEAT (100 mg/kg) significantly alleviated OVX-induced expanded size of adipose tissue as well as steatosis in non-adipose tissues. Previously, it was reported that the ethanol extract of *A. subulatum* inhibits hepatic steatosis through antioxidant activity or by decreasing infiltration of inflammatory cells in obesity-induced animal models [17,28]. *A. subulatum* also inhibits adipogenesis by reducing PPARγ and C/EBPα expression in 3T3-L1 cells [18]. It is known that adipocytes increase the levels of inflammatory adipokines to recruit more inflammatory cells that stimulate the obesity state [29,30]. In overweight and obese women, elevated serum levels of pro-inflammatory cytokines (IL-6 and TNF-α) are generally observed [31]. Thus, the significant inhibition of hepatic steatosis and adipocyte enlargement in OVX mice by EEAT (100 mg/kg) might have been due to its anti-inflammatory and anti-adipogenic activity. It is known that PPARγ agonists attenuate adipogenesis but cannot promote osteogenesis in bone marrow mesenchymal cells [32,33]. As EEAT decreases fat accumulation in bone marrow and *A. subulatum* regulates PPARγ expression, it should be further elucidated whether EEAT might regulate osteogenic or adipogenic lineage from bone marrow mesenchymal cells.

### 2.4. Phytochemical Profiling of EEAT

To elucidate the molecular basis of its pharmacological actions, we investigated the phytochemical profile of EEAT through spectrometry and comparison of absorbance and mass fragmentation of standard marker components; UHPLC–MS/MS analysis identified two phenolic components (p-hydroxybenzoic acid and vanillin) and six flavonoids (epicatechin, rutin, isoquercitrin, taxifolin, (+)-hannokinol, and quercetin), as shown in Table 1. UV chromatograms with different spectra of individual components are shown in Figure 5. It has been reported that some components (rutin, isoquercitrin, taxifolin, and quercetin) have an inhibitory effect on OVX-induced osteoporosis without estrogenic effects on the uterus [34,35,36]. In addition, they also inhibited osteoclast differentiation by decreasing reactive oxygen species or osteoclast activity by decreasing inflammatory cytokine levels [37,38]. Epicatechin, rutin, isoquercitrin, and quercetin are involved in lipid or energy metabolism through SREBP/AMPK signaling or the fat oxidation pathway [39,40,41]. It has been reported that rutin (quercetin-3-*O*-β-rutinoside) and isoquercitrin (quercetin-3-*O*-β-glucoside) are modified glycoside forms of quercetin, known as phytoestrogen, but have different bioavailability in vivo [42]. Thus, it may be suggested that the pharmacological effect of EEAT against osteoporosis or obesity results from the cumulative effect of the active components rather than their individual activity. A study assessing the adequate concentration and bioavailability of individual active components to enhance bone health or obesity should be performed in the future.

## 3. Materials and Methods

### 3.1. Materials

p-hydroxybenzoic acid, vanillin, epicatechin, rutin, isoquercitrin, taxifolin, (+)-hannokinol, and quercetin were purchased from Targetmol (Boston, MA, USA). All chemicals and solutions for HPLC analysis were MS grade from Thermo Fisher Scientific (Rockford, IL, USA). *Amomum tsao-ko* was purchased from the National Development Institute of Korean Medicine (Gyeongsan, Korea). After ethanol extraction, the EEAT was concentrated using a vacuum concentrator and freeze dryer, and then stored at −20 °C.

### 3.2. Osteoclast Culture and Bone Resorption Assay

Bone marrow cells were harvested from mouse long bones as reported previously [43]. BMMs were obtained from the bone marrow using the noncoating plate attachment method and cultured in α-Minimum Essential Medium (Thermo Fisher Scientific) supplemented with 10% fetal bovine serum, 1% penicillin/streptomycin, and macrophage colony-stimulating factor (M-CSF) (60 ng/mL) as described previously [44]. RANKL (100 ng/mL, four days) was used for the induction of osteoclast differentiation in the above culture medium. Among TRAP-stained cells, multinuclear cells with gigantic morphology were enumerated as mature osteoclasts. Cell toxicity or proliferation were examined by measuring the generated amount of formazan dye (Cell Counting Kit-8) (Dojindo, Japan) after a two-day cultivation of BMMs with the vehicle (0.1% dimethyl sulfoxide in above medium) or indicated concentration of EEAT. Bone resorption activity of mature osteoclasts was examined by the generation of a resorption pit on the artificial bone surface coated with hydroxyapatite (Osteo Assay Surface plates, Corning, New York, NY, USA). Mature osteoclasts were generated on collagen gel as described previously [43], placed on the artificial bone surface, and treated with the vehicle or EEAT for 16 h. Cells were washed out with 2.5% sodium hypochlorite, and then generated resorption pits were photographed for image analysis with Image J software (National Institutes of Health, Bethesda, Rockville, MD, USA).

### 3.3. Quantitative Real-Time Polymerase Chain Reaction (RT-PCR)

Total RNA having 18S rRNA was purified using silica-membrane-based spin columns (Qiagen, Hilden, Germany). Total RNA (1 μg) was incubated with reverse transcriptase, oligo-dT primer, deoxynucleoside triphosphate, and RT buffer to make cDNA (Applied Biosystems, Foster City, CA, USA). Quantification of mRNA expression was evaluated by a two-step PCR reaction with a specific TaqMan primer, cDNA, and the TaqMan Universal PCR Master Mix (Applied Biosystems) on the ABI QuantStudio 6 Flex RT-PCR system. Specific primers for each target gene were c-Fos (Mm00487425_m1), NFATc1 (Mm00479445_m1), DC-STAMP (Mm01168058_m1), Atp6v0d2 (Mm00656638_m1), and GAPDH (Mm99999915_g1) (Applied Biosystems). All qPCR reactions were the average of triplication. The comparative delta Ct method was used for the calculation of the relative expression of the target genes with GADPH as internal control gene. Data were expressed as a relative fold change compared to the M-CSF-treated control. Three time-independent experiments were performed, and data from representative experiment were shown.

### 3.4. Western Blot

Total proteins were obtained using RIPA buffer by repeated vortexing and incubation in ice. Protease and a phosphatase inhibitor cocktail were added to prevent protein degradation and de-phosphorylation (Thermo Fisher Scientific). After centrifugation, the total soluble protein was quantified by detection of the cuprous cation by bicinchoninic acid (Pierce Biotechnology, Rockford, IL, USA). Protein separation based on its molecular weight was performed by SDS-PAGE Precast Protein Gels (Bio-rad, Hercules, CA, USA). Protein on the gels was transferred onto polyvinylidene fluoride membranes by semi-dry blotting protocol. Subsequently, the membranes were immunoblotted with primary antibodies specific to the target protein. All primary antibodies were from Cell Signaling Technology (Danvers, MA, USA) except c-Fos and NFATc1 (Santa Cruz Biotechnology, Dallas, TX, USA). The secondary antibody was horseradish peroxidase. Conjugated secondary antibodies reacted with ECL Western blotting substrate (Bio-Rad). Representative development images of Western blot were acquired under LAS-4000 image analyzer (Fujifilm, Tokyo, Japan).

### 3.5. Animal Study

Female C57BL/6J mice (SLC Inc., Shizuoka, Japan) weighing 19–20 g were housed with standard chow diet and water ad libitum in a specific pathogen-free (SPF) animal facility of Knotus (Guri, Korea). SPF facility was maintained in the standard laboratory condition of temperature (22 °C ± 2 °C), humidity (55 ± 5%), and illumination circle (12 h light/dark cycle). After one-week adaptation in SPF environment, 6-weeks old mice were bilaterally ovariectomized or sham-operated through a dorsal approach. Healthy mice that recovered from OVX surgery were selected and randomly assigned into four groups (n = 6). Sham and OVX group were orally administered the vehicle. low concentration EEAT (EEAT-L) and high concentration EEAT (EEAT-H) groups were orally administered EEAT 30 and 100 mg/kg/day, respectively. The mice were fed with a commercial low-fat diet with 10 kcal% fat (Research Diets, New Brunswick, NJ, USA). Animal study proposals containing aim and protocols were reviewed and approved by the Animal Care and Use Committee of Knotus.

### 3.6. Micro-CT Analysis

SkyScan, model 1276 µCT scanner (Bruker, Kontich, Belgium) was applied to evaluate microstructural changes in the distal femur. After scanning the femur, an image reconstruction or analysis was performed using the SkyScan NRecon program or SkyScan CTAn software. The starting point of the volume measurement was set at 80 µm from the lower end of the growth plate and reached to 1.2 mm high with 150 cross-sections. Five bone morphometric parameters, including trabecular bone mineral density (BMD, g/cm^3^), trabecular bone volume fraction (BV/TV, %), trabecular number (Tb.N, 1/mm), trabecular separation (Tb.Sp, µm), and trabecular thickness (Tb.Th, µm), were calculated as per the instruction of program.

### 3.7. Measurement of Bone Turnover Markers

Serum levels of CTX and PINP were measured using ELISA kits (Immunodiagnostic Systems Ltd., London, UK).

### 3.8. Histological Analysis

The tissues isolated were fixed using 10% neutral buffered formalin for 48 h at room temperature. After dehydration using a series of ethanol (70–100%), tissues were embedded in paraffin to obtain a histological section block. RDO Gold (RDO, Aurora, IL, USA) was used to decalcify bone tissue for 1 week in between fixation and dehydration. Paraffin-embedded tissues were stained with hematoxylin and eosin. Fat size and lipid droplet area in representative images were obtained under light microscopy and analyzed using the Image J software (National Institutes of Health, Bethesda, Rockville, MD, USA).

### 3.9. UHPLC–MS/MS

Dionex UltiMate 3000 system equipped with a Thermo Q-Exactive mass spectrometer was applied to analysis EEAT. Xcalibur v.3.0 and Tracefinder v.3.2 software was used to acquire analysis data and control the system. Chromatographic separation was performed with a C18 column (Acquity BEH, 100 × 2.1 mm, 1.7 μm) at 35 °C. Operation procedure and the condition of UHPLC or mass spectrometer were referenced to in the previous study with some modifications [45].

### 3.10. Statistical Analysis

In vivo or in vitro data were presented as mean and standard error or standard deviation, respectively. A data comparison of the two groups was performed by a one-way analysis of variance with Dunnett’s post hoc test. Multiple comparisons of more than two groups were analyzed by a two-way ANOVA followed by Bonferoni’s post hoc test in GraphPad Prism version 8 (San Diego, CA, USA). Results were considered statistically significant when *p* values were less than 0.05.

## 4. Conclusions

This study is the first of its kind to evaluate EEAT inhibitory activity on OVX-induced bone loss. In particular, the inhibitory effect of EEAT on osteoclastogenesis through RANKL-induced NF-κB/Fos/NFATc1 pathway appears to play a crucial role in preventing OVX-induced bone loss. Additionally, suppression by EEAT of OVX-induced body weight gain and fat accumulation in adipose, liver, and bone marrow tissues suggests its regulatory effect on estrogen deficiency-induced adipogenesis in menopause. Furthermore, we identified eight phytochemical components in EEAT that are known to have an inhibitory effect on trabecular bone loss or fat accumulation in the liver or adipocytes. Taken together, these findings suggest that EEAT is an effective nutraceutical candidate that works by altering osteoporosis and lipid metabolism to attenuate obesity during the menopausal state.

## Figures and Tables

**Figure 1 molecules-26-00784-f001:**
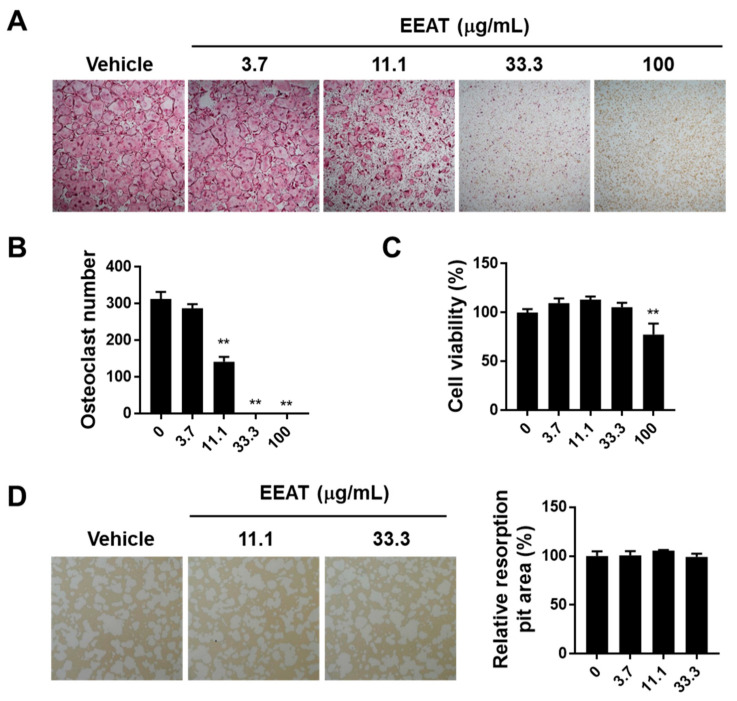
Ethanol extract of *Amomum tsao-ko* (EEAT) inhibits osteoclastogenesis in bone marrow-derived macrophage cells (BMMs) stimulated with receptor activator of nuclear factor-kappa Β ligand (RANKL). (**A**) BMMs were differentiated to osteoclasts in the presence of macrophage colony-stimulating factor (M-CSF) and RANKL for four days. Vehicle or EEAT were treated in the culture medium at day 0. Generated osteoclasts were stained with tartrate-resistant acid phosphatase (TRAP) staining solution and photographed under a microscope (×4). (**B**) After TRAP staining, multinuclear osteoclasts were enumerated under the microscope (×4). (**C**) BMMs were incubated with M-CSF and the indicated concentrations of EEAT for 24 h and then assayed with CCK-8 solution to measure cell viability. (**D**) Mature osteoclasts were incubated with EEAT on the artificial bone surface to evaluate its effect on resorption activity. Resorption pits were photographed and analyzed. ** *p*  <  0.01 versus vehicle.

**Figure 2 molecules-26-00784-f002:**
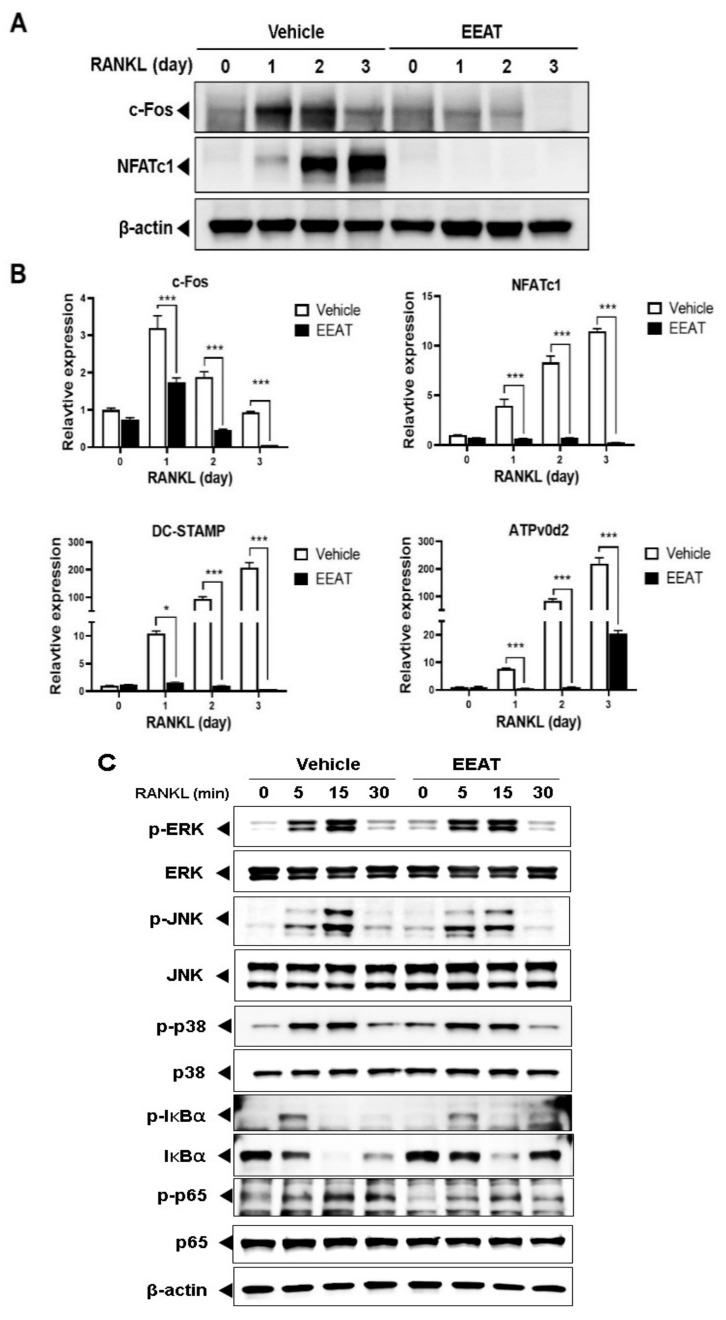
Ethanol extract of *Amomum tsao-ko* (EEAT) inhibits receptor activator of nuclear factor-kappa Β ligand (RANKL)-induced NF-κB/Fos/NFATc1 signaling pathway. Bone marrow-derived macrophage cells (BMMs) were pre-incubated with the vehicle or EEAT (33.3 µg/mL) for 3 h and then simulated with RANKL (50 ng/mL) for four days. (**A**) The protein levels of c-Fos, nuclear factor of activated T-cells cytoplasmic 1 (NFATc1), and β-actin were determined by Western blot with specific antibodies. (**B**) mRNA expression levels of c-Fos, NFATc1, dendritic cell–specific transmembrane protein (DC-STAMP), and ATPv0d2 were examined by Real-Time Polymerase Chain Reaction (RT-PCR). Day 0 represents the day of plating BMMs untreated with RANKL. Artificial fold change from day 0 to day 3 were represented in comparison with RANKL-untreated vehicle. * *p*  <  0.05 or *** *p*  <  0.01 versus vehicle. (**C**) BMMs were pre-incubated with EEAT for 3 h and then stimulated with RANKL for 5, 15, and 30 min. Protein levels of each target were analyzed by Western blot analysis with the indicated antibodies.

**Figure 3 molecules-26-00784-f003:**
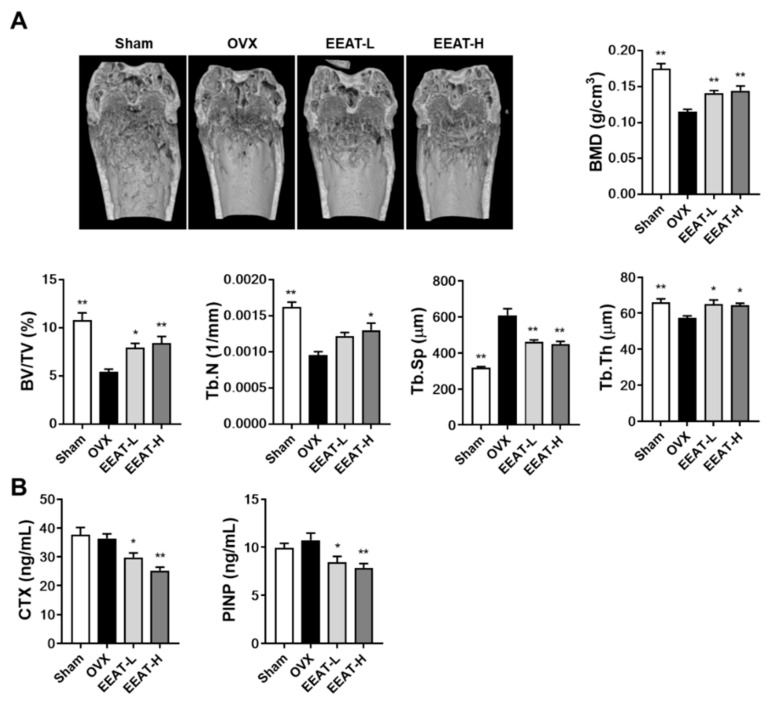
Ethanol extract of *Amomum tsao-ko* (EEAT) inhibits ovariectomy (OVX)-induced trabecular loss. After oral administration of the vehicle, low concentration EEAT (EEAT-L, 30 mg/kg), or high concentration EEAT (EEAT-H, 100 mg/kg) for five weeks, mouse right femur and serum were collected. (**A**) The femur was scanned and analyzed by micro-CT. The radiological images were constructed from distal femora. Five bone morphometric parameters were analyzed in the trabecular area below the lower end of the growth plate: bone mineral density (BMD), trabecular bone volume fraction (BV/TV), trabecular number (Tb.N), trabecular separation (Tb.Sp), and trabecular thickness (Tb.Th). (**B**) Serum levels of C-terminal telopeptide of type I collagen (CTX) and procollagen type I N-terminal propeptide (PINP) were measured. * *p*  <  0.05, ** *p*  < 0.01 versus OVX group.

**Figure 4 molecules-26-00784-f004:**
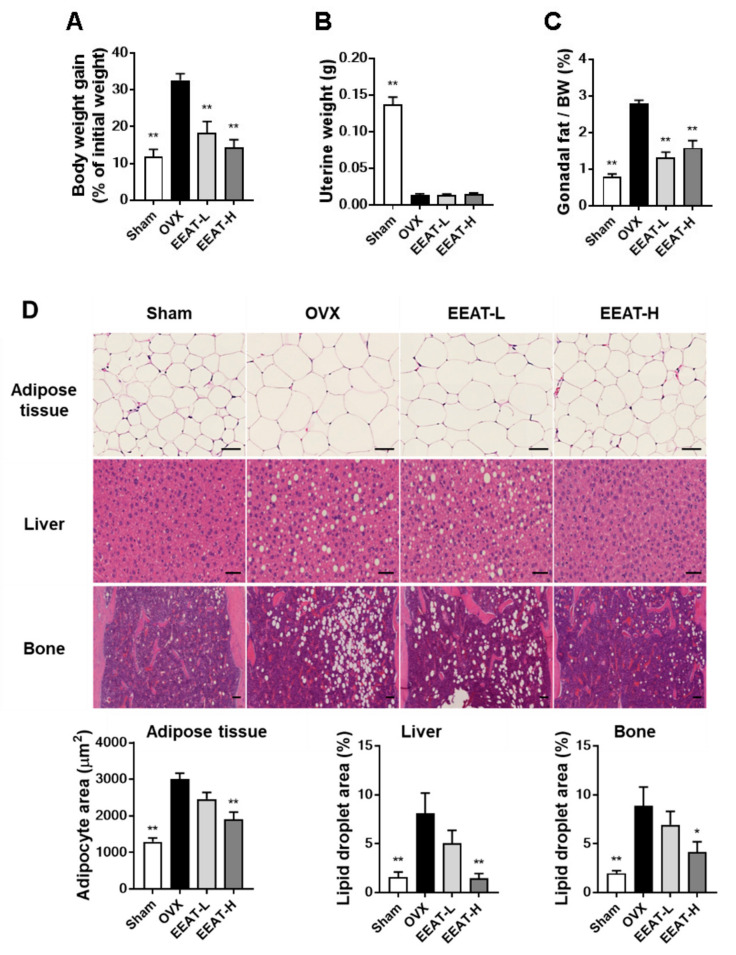
Ethanol extract of *Amomum tsao-ko* (EEAT) attenuates ovariectomy (OVX)-induced fat accumulation. Sham or OVX mice were orally received the vehicle or different doses of EEAT (30 or 100 mg/kg). After five weeks of administration, (**A**) body weight gain during EEAT administration period was assessed. Mice were euthanized, and then weight change of (**B**) uterus or (**C**) gonadal fat was quantified. (**D**) Isolated tissues were fixed and stained by hematoxylin and eosin staining solution (scale bar, 50 µm). Histological image of the adipocyte area or lipid droplets in each tissue was analyzed, and representative images are shown. * *p*  <  0.05, ** *p*  < 0.01 versus OVX group.

**Figure 5 molecules-26-00784-f005:**
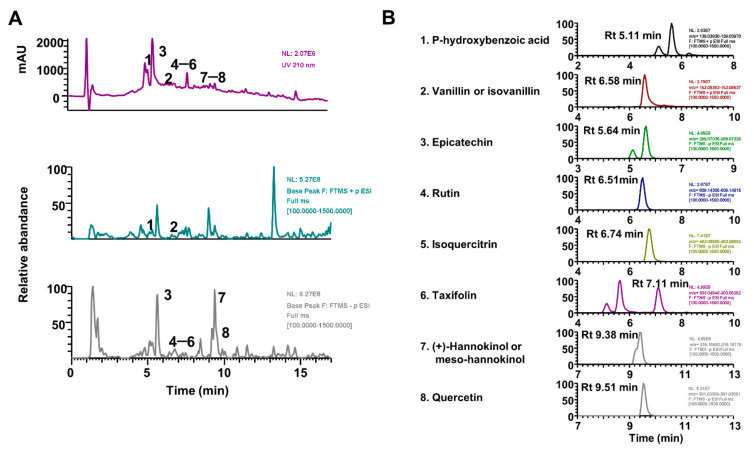
Ultrahigh-performance liquid chromatography–tandem mass spectrometry (UHPLC–MS/MS) chromatograms of ethanol extract of Amomum tsao-ko (EEAT). (**A**) Ultraviolet chromatograms at 210 nm and base peak chromatograms with positive and negative modes of EEAT. (**B**) Individual ion chromatograms of eight components in EEAT.

**Table 1 molecules-26-00784-t001:** List of identified phytochemicals in ethanol extract of *Amomum tsao-ko* (EEAT) by ultrahigh-performance liquid chromatography-tandem mass spectrometry (UHPLC–MS/MS) analysis.

No	R_t_ (Min)	Calculated (*m*/*z*)	Estimated (*m*/*z*)	Adducts	Error (ppm)	Formula	MS/MS Fragments (*m*/*z*)	Identifications
1	5.06	139.0390	139.0388	[M + H]^+^	−1.5296	C_7_H_6_O_3_	-	p-Hydroxybenzoic acid *
2	6.52	153.0546	153.0543	[M + H]^+^	−1.852	C_8_H_8_O_3_	153.0542, 125.0595, 93.0338	Vanillin or isovanillin *
3	5.64	289.0718	289.0723	[M − H]^−^	1.8547	C_15_H_14_O_6_	289.0721, 245.0818, 203.0710, 179.0342	Epicatechin *
4	6.53	609.1461	609.1472	[M − H]^−^	1.817	C_27_H_30_O_16_	609.1463, 301.0349, 300.0278	Rutin *
5	6.79	463.0882	463.0890	[M − H]^−^	1.7702	C_21_H_20_O_12_	463.0888, 301.0348, 300.0278	Isoquercitrin *
6	7.12	303.0510	303.0515	[M − H]^−^	1.596	C_15_H_12_O_7_	285.0408, 177.0185, 125.0231	Taxifolin *
7	9.48	315.1602	315.1606	[M − H]^−^	1.554	C_19_H_24_O_4_	315.1605, 149.0597	Meso-hannokinol or (+)-hannokinol
8	9.58	301.0354	301.0358	[M − H]^−^	1.3856	C_15_H_10_O_7_	301.0356, 178.9977, 151.0026	Quercetin *

* Comparison with the retention time (Rt) and MS spectral data of authentic standards.

## Data Availability

The data presented in this study are available on request from the corresponding author.

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
