# Peer review of "Ethanol Extract of Amomum tsao-ko Ameliorates Ovariectomy-Induced Trabecular Loss and Fat Accumulation"

_molecules, 2021, doi:10.3390/molecules26040784_

Round 1
Reviewer 1 Report
Dear authors,
I have read the manuscript molecules-1067753 thoroughly. The submitted manuscript presents the results of a study performed with ethanolic extract of Amomum tsao-ko on cell cultures and female mice. Overall I believe the quality of the presentation is currently inadequate for the paper to be published in this magazine. There are numerous limitations to the study described in the paper. The most serious drawback is that much information is missing about the methods used in a study. As a result, all the work presented becomes fragile.
Introduction is the best part of the manuscript and gives all the necessary information for readers. There is only one stylistic mistake in line 44 that should be rewritten.
“Methods and materials” should include all the necessary data to enable the reader to follow the described procedures. For example in line 214 dimethyl sulfoxide was used as a vehicle for two-day cultivation of BMMs. I suppose that the mixture with aqueous buffer was used. What was the concentration on DMSO, which, if any buffer was used? Additionally in line 217 sodium hydrochlorite is mentioned. Probably it should be replaced with hypochlorite. Again concentration is missing. Similar “black holes” are throughout this part of the text.
In animal studies the administration of EEAT is described (Lines 253-254). How was EEAT administered? Only if EEAT was added orally, the conclusions about nutraceutical candidates (Lines 293-295, abstract) are supported, since otherwise it is hard to speak about absorption of active compounds. I am not certain that was the case in your work.
Due to the inferior presentation of methods the “Results and Discussion” part is somewhat questionable. We know a lot about in vitro effects of phenolic antioxidants. In a way this study was presented, the paper lacks two important features for publication in a high ranking magazine: novelty and interest for the readers. I would suggest the authors completely rewrite the manuscript, starting with the precise description of methods and resubmit the article.
Kind regards,
Author Response
- Introduction is the best part of the manuscript and gives all the necessary information for readers. There is only one stylistic mistake in line 44 that should be rewritten.
Re) Thanks for the comments. The relevant statement has been modified as follows (lines 43-44) :
“Estrogen is shown to exert a regulatory effect on food intake and energy expenditure by interacting with neuropeptides in rodent models [7]”
- “Methods and materials” should include all the necessary data to enable the reader to follow the described procedures. For example in line 214 dimethyl sulfoxide was used as a vehicle for two-day cultivation of BMMs. I suppose that the mixture with aqueous buffer was used. What was the concentration on DMSO, which, if any buffer was used? Additionally in line 217 sodium hydrochlorite is mentioned. Probably it should be replaced with hypochlorite. Again concentration is missing. Similar “black holes” are throughout this part of the text.
Re) Thank you for pointing out the error. We have changed “dimethyl sulfoxide” to “0.1% dimethyl sulfoxide in above medium” (line 220). In addition, the word “sodium hydrochlorite” has been collected to “2.5% sodium hypochlorite” (line 224).
- In animal studies the administration of EEAT is described (Lines 253-254). How was EEAT administered? Only if EEAT was added orally, the conclusions about nutraceutical candidates (Lines 293-295, abstract) are supported, since otherwise it is hard to speak about absorption of active compounds. I am not certain that was the case in your work.
Re) Mice were orally administered with EEAT. The relevant statement has been modified as follows (lines 260-261):
“EEAT-L and EEAT-H groups orally administered with EEAT 30 and 100 mg/kg/day, respectively.”
- Due to the inferior presentation of methods the “Results and Discussion” part is somewhat questionable. We know a lot about in vitro effects of phenolic antioxidants. In a way this study was presented, the paper lacks two important features for publication in a high ranking magazine: novelty and interest for the readers. I would suggest the authors completely rewrite the manuscript, starting with the precise description of methods and resubmit the article.
Re) We have provided additional data and precise description of methods in the revised version. In the present study, we first showed the novel dual effects of EEAT (Ethanol extract of Amomum tsao-ko) on menopausal osteoporosis and obesity, that may be of interest to readers.
Reviewer 2 Report
The manuscript is interesting and the scientific design is well structured. The biotechnologies used are varied and detailed. Phytochemical components of EEAT could be useful in the prevention of osteoporosis and metabolic alterations that accompany menopause
Author Response
The manuscript is interesting and the scientific design is well structured. The biotechnologies used are varied and detailed. Phytochemical components of EEAT could be useful in the prevention of osteoporosis and metabolic alterations that accompany menopause
Re) Thank you for your comments.
Reviewer 3 Report
The paper is interesting and well witten, but I have some concerns:
Major:
- The authors reported that 33.3 μg/ml EEAT inhibited osteoclastogenesis, thus I think that is not possible that at this concentration the resorption area is elevated or not different respect to the control condition.
- Images of resorption area should be provided
- Figure 2B: graphs should be improved
- in in vivo experiments histology of bone should be reported with osteoclast and osteoblast counts
- Figure 4D: adipose tissue images should be improved
Minor:
The following papers should be reported in the introduction or discussion:
Front Immunol. 2019 May 3;10:1001. doi: 10.3389/fimmu.2019.01001.
J Pathol. 2020 Apr;250(4):440-451.
Author Response
- The authors reported that 33.3 μg/ml EEAT inhibited osteoclastogenesis, thus I think that is not possible that at this concentration the resorption area is elevated or not different respect to the control condition.
- Re) As shown in Figure 1A and 1D, EEAT (33.3 μg/ml) inhibited RANKL-induced differentiation of BMMs to osteoclasts, but did not affect resorbing activity of mature osteoclasts. To avoid confusion, the relevant paragraph in the Materials and Methods section has been rephrased as follows (lines 221-226):
“Bone resorption activity of mature osteoclasts was examined by generation of resorption pit on artificial bone surface coated with hydroxyapatite (Osteo Assay Surface plates, Corning, NY, USA). Mature osteoclasts were generated on collagen gel as described previously [43], placed on artificial bone surface, and treated with vehicle or EEAT for 16 h. Cells were washed out by 2.5% sodium hypochlorite, and then generated resorption pits were photographed for image analysis with Image J software (National Institutes of Health, Bethesda, MD, USA).”
- Images of resorption area should be provided
Re) As recommended, the images have been provided in Figure 1D.
- Figure 2B: graphs should be improved
- Re) As recommended, the graphs have been improved.
- in in vivo experiments histology of bone should be reported with osteoclast and osteoblast counts
- Re) Thank you for your comments. To address the reviewer’s concerns, we have included the data showing the effects of EEAT on serum levels of CTX and PINP as bone turnover markers in Figure 3B, and the relevant paragraph has been modified as follows (lines 127-133):
“To further investigate the effect of EEAT on bone metabolism, serum bone turnover markers were measured. EEAT decreased both the levels of C-terminal telopeptide of type I collagen (CTX), a bone resorption marker, and procollagen type I N-terminal propeptide (PINP), a bone formation marker, compared with the OVX group (Figure 3B). Therefore, these results suggest that improvement of the bone structure parameters in the OVX model by EEAT administration could have resulted from its effect to decrease the turnover rate of bone remodeling at least in part by suppressing osteoclast-mediated bone resorption.”
- Figure 4D: adipose tissue images should be improved
Re) As recommended, the images have been improved in Figure 4D.
Minor:
The following papers should be reported in the introduction or discussion:
Front Immunol. 2019 May 3;10:1001. doi:10.3389/fimmu.2019.01001.
J Pathol. 2020 Apr;250(4):440-451.
Re) As recommended, the papers have been cited in the revised manuscript (lines 58-59, line 213-215).
Round 2
Reviewer 1 Report
Dear authors,
Thank you for the corrections you made to the manuscript. I will suggest the editor to accept the paper for publication.
Kind regards.
Reviewer 3 Report
The authors addressed my concerns